# Vasodilatory Effect of *n*-Butanol Extract from *Sanguisorba officinalis* L. and Its Mechanism

**DOI:** 10.3390/plants14071095

**Published:** 2025-04-01

**Authors:** Hangyu Jin, Jiaze Li, Shuyuan Wang, Enyi Jin, Jun Zhe Min, Gao Li, Yun Jung Lee, Lihua Cao

**Affiliations:** 1Clinical Medicine, College of Medicine, Yanbian University, Yanji 133002, China; 18643337786@163.com; 2Key Laboratory of Natural Medicines of the Changbai Mountain, Ministry of Education, Medical Pharmacy, Yanbian University, Yanji 133002, China; 2023010938@ybu.edu.cn (J.L.); 17289031163@163.com (E.J.); junzhemin23@163.com (J.Z.M.); gli@ybu.edu.cn (G.L.); 3Center of Morphological Experiment, College of Medicine, Yanbian University, Yanji 133002, China; 15765126992@163.com; 4Wonkwang-Oriental Medicines Research Institute, Wonkwang University, Iksan 54538, Republic of Korea

**Keywords:** *Sanguisorba officinalis*, Diyu, vasorelaxation, NO-cGMP, K^+^ channels, PI3K-Akt

## Abstract

The dried root of *Sanguisorba officinalis* L. (commonly known as Diyu) has been studied for its various pharmacological effects, including its antibacterial, antitumor, antioxidant, and anti-inflammatory activities. In the present study, primary cultured vascular endothelial cells (HUVECs) and isolated phenylephrine-precontracted rat thoracic aortic rings were examined to investigate the possible mechanism of a butanol extract of Diyu (BSO) in its vascular relaxant effect. HUVECs treated with BSO produced a significantly higher amount of nitric oxide (NO) compared to the control. However, its production was inhibited by pretreatment with N^G^-nitro-L-arginine methylester (L-NAME) or wortmannin. BSO also increased the phosphorylation levels of endothelial nitric oxide synthase (eNOS) and Akt. In the aortic ring, BSO relaxed PE-precontracted rat thoracic aortic rings in a concentration-dependent manner. The absence of the vascular endothelium significantly attenuated BSO-induced vasorelaxation. The non-selective NOS inhibitor, L-NAME, and the selective inhibitor of soluble guanylyl cyclase (sGC), 1H-[1,2,4]-oxadiazolo-[4,3-α]-quinoxalin-1-one (ODQ), dramatically inhibited the BSO-induced relaxation effect of the endothelium-intact aortic ring. Ca^2+^-free buffer and intracellular Ca^2+^ homeostasis regulators (TG, Gd^3+^, and 2–APB) inhibited BSO-induced vasorelaxation. In Ca^2+^-free Krebs solution, BSO markedly reduced PE-induced contraction. Vasodilation induced by BSO was significantly inhibited by wortmannin, an inhibitor of Akt. Pretreatment with the non-selective inhibitor of Ca^2+^-activated K^+^ channels (K_Ca_), tetraethylammonium (TEA), significantly attenuated the BSO-induced vasorelaxant effect. Furthermore, BSO decreased the systolic blood pressure and heart rate in a concentration-dependent manner in rats. In conclusion, BSO induces vasorelaxation via endothelium-dependent signaling, primarily through the activation of the PI3K-Akt-eNOS-NO signaling pathway in endothelial cells, and the activation of the NO-sGC-cGMP-K⁺ channels pathway in vascular smooth muscle cells. Additionally, store-operated Ca^2+^ entry (SOCE)-eNOS pathways and the inhibition of Ca^2^⁺ mobilization from intracellular stores contribute to BSO-induced vasorelaxation.

## 1. Introduction

The vascular endothelium is located in the vascular wall between the lumen and the smooth muscle cells and plays a critical role in regulating homeostatic functions within the vasculature. This endothelial cell layer is capable of conducting blood signals, sensing mechanical forces within the lumen, and producing a variety of factors that influence vascular function [1,2]. The vascular endothelium produces potent vasodilators, such as endothelium-derived relaxing factors (EDRFs), prostacyclin, and endothelium-derived hyperpolarizing factors. Among these, the EDRF, which has been identified as nitric oxide (NO), is considered the most crucial vasodilator in the endothelium [3]. Endothelium-derived NO is considered to be a pleiotropic biological mediator that regulates a wide variety of activities, ranging from neural function to vasoreactivity [4]. NO has been identified as a neurotransmitter in both the peripheral and central nervous system [5]. NO mediates numerous responses within the autonomic and cardiovascular systems and plays a role in the regulation of blood flow and blood pressure [6], and in the inhibition of urethral relaxation during the gastrointestinal power micturition reflex [7]. Specifically, in vascular smooth muscle cells, NO activates soluble guanylate cyclase, which catalyzes the formation of the second messenger cGMP, leading to vasodilation [8,9]. In addition to its vasodilatory effects, NO inhibits smooth muscle cell contraction, migration, and proliferation, as well as the production of endothelin, platelet aggregation, and the infiltration of leukocytes into the endothelium, thereby preventing the development of atherosclerosis [10,11]. Furthermore, L-arginine supplementation in healthy individuals with essential hypertension has been shown to induce a rapid decrease in both systolic and diastolic blood pressure [12].

The dried root of *Sanguisorba officinalis* L., commonly known as Diyu, belongs to the Rosaceae family. Diyu has a sour and bitter taste and is characterized by a cool nature. It is widely distributed across northern China, Guangxi, and Europe. Preferring sandy soil, Diyu is typically found on sunny slopes, in forests, and in valleys. Furthermore, Diyu is highly adaptable, being both cold-resistant and drought-tolerant, enabling it to sprout new leaves in the spring, summer, and autumn. Recent studies have highlighted the novel pharmacological effects of Diyu, including its antibacterial, antitumor, and antioxidant activities. Additionally, preparations derived from Diyu have been widely applied in clinical settings for the rapid and effective treatment of various inflammations, for the prevention and treatment of bone marrow suppression induced by tumor chemotherapy, and for the effective treatment of uterine fibroids [13]. Current research, both domestic and international, on the chemical constituents of Diyu has primarily focused on the major components found in its rhizomes: tannins, phenolic acids, saponins, flavonoids, and polysaccharides [14]. Recently, it has been reported that the ethanol extract of Diyu exhibits hypotensive and vasorelaxant effects [15]. However, studies on the vasodilatory effects of the n-butanol extract of Diyu (BSO) remain insufficiently comprehensive and detailed. This study aims to investigate the effects and underlying mechanisms of BSO on nitric oxide’s (NO) production and vasodilation pathways using isolated thoracic aorta from Sprague Dawley rats and human umbilical vein endothelial cells (HUVECs).

## 2. Results

### 2.1. Results of HPLC-DAD and UPLC-MS for the Identification of Diyu Components

The BSO was processed using the Xcalibur 4.0 workstation, and Compound Discoverer 3.2 software was used to obtain the matched peak data from the UPLC-MS raw data. The results of the HPLC-DAD determination and the qualitative analysis by UPLC-MS of the *n*-butanol extract of Diyu are shown in Figure 1. The results were processed to show the following: there were 23 chemical components in the *n*-butanol extract, with the amino acid compounds mainly including L-arginine, D-proline, and D(-)-2-aminobutyric acid; the nucleotide compounds mainly included adenosine; the flavonoids mainly included catechin and dihydroquercetin; and the polyphenolic compounds mainly included gallic acid as well as the alkaloidal constituents of oxymatrine, natural antioxidant ellagic acid, and other chemical constituents. The UPLC-MS putative compounds are shown in Table 1.

### 2.2. Effect of BSO on NO Production in HUVECs

In order to observe the effect of BSO on the survival rate of HUVECs, we used the MTT method. The BSO concentrations of 10 μg/mL and 50 μg/mL used in this experiment had no cytotoxic effect on HUVECs. The fluorescence intensity of adherent HUVECs was examined by fluorescence microscopy. NO production was significantly increased by BSO (10, 50 μg/mL) treatment compared to normal HUVECs (BSO-untreated group) (Figure 2). The NO synthase inhibitor L-NAME (1 × 10^−4^ mol/L) and the PI3K inhibitor wortmannin (1 × 10^−7^ mol/L) significantly decreased BSO-induced NO production (*p* < 0.01). BSO-induced NO production was significantly inhibited by L-NAME and wortmannin, suggesting that it was mediated by the Akt/NO pathway.

### 2.3. Effects of BSO-Induced eNOS and Akt Phosphorylation in HUVECs

To further characterize the involvement of BSO-activated eNOS in vasodilatation, HUVECs were utilized to observe the changes in the PI3K/Akt pathway. The phosphorylation levels of eNOS (140 kDa) and Akt (56 kDa) were determined by Western blot. BSO concentration-dependently increased Akt and eNOS phosphorylation levels. However, it was significantly inhibited by the PI3K/Akt inhibitor wortmannin (1 × 10^−7^ mol/L) (Figure 3), suggesting that the PI3k/Akt pathway mediates BSO-induced eNOS phosphorylation.

### 2.4. Effect of BSO on Tension in Endothelium-Intact and Endothelium-Removed Vascular Rings

BSO (1–100 μg/mL) concentration-dependently induced vasorelaxation in endothelium-intact aortic rings precontracted with PE (1 × 10^−6^ mol/L), whereas the vasodilatory response of BSO was significantly inhibited after endothelium removal (*p <* 0.001) (Figure 4A). This indicated that the vasodilatory effect of BSO on blood vessels was endothelium-dependent.

### 2.5. Effects of L-NAME and ODQ on BSO-Induced Vasodilatory Effects

Because the NO-cGMP signaling pathway in endothelial cells has an important role in regulating endothelium-dependent vasodilation, endothelium-intact vascular rings pretreated with the guanylate cyclase inhibitor ODQ (1 × 10^−5^ mol/L) and the NO synthase inhibitor L-NAME (1 × 10^−5^ mol/L) markedly inhibited the diastolic effect of BSO (*p <* 0.001) (Figure 4B). This suggests that BSO has a diastolic effect through the NO-sGC-cGMP signaling pathway to exert vasodilatory effects.

### 2.6. Effect of Akt on BSO-Induced Vasodilatory Effect

To investigate whether the BSO-induced vasodilatory effect was related to non-calcium-dependent Akt signaling, endothelium-intact vascular rings were pretreated with the Akt inhibitor wortmannin (1 × 10^−7^ mol/L) to markedly inhibit the BSO-induced vasodilatory effect (*p <* 0.001) (Figure 4C). This indicates that BSO exerts vasodilatory effects through activating the Akt-eNOS-cGMP signaling pathway.

### 2.7. Effect of Extracellular Ca^2+^ on BSO-Induced Vasodilatory Effect

To investigate whether the BSO-induced vasodilatory effect was related to extracellular Ca^2+^ influx, Ca^2+^-free Krebs solution inhibited the BSO-induced vasodilatory effect (*p* < 0.01), whereas diltiazem (1 × 10^−5^ mol/L), an L-type voltage-dependent calcium channel inhibitor, had no effect (Figure 4D). This suggests that the Ca^2+^ required for BSO-induced vasodilation does not influx through L-type channels but is accomplished through other pathways.

### 2.8. Effects of SOCE Modulators on BSO-Induced Vasodilatory Effects

To further confirm the Ca^2+^-dependent vasodilatory effect of BSO, the store-operated Ca^2+^ entry (SOCE) modulator Gd^3+^ (1 × 10^−5^ mol/L) was able to inhibit the BSO-induced vasodilatory effect (*p* < 0.05), and TG (1 × 10^−6^ mol/L) and 2-APB (7.5 × 10^−5^ mol/L) significantly inhibited the BSO-induced vasodilatory effect (*p* < 0.01) (Figure 5A). This indicated that SOCE was involved in the BSO-induced vasodilatory effect.

### 2.9. Effect of Potassium Channels on BSO-Induced Vasodilatory Effects

To determine whether K^+^ channels were involved in the vasodilatory effects induced by BSO, endothelium-intact vessels were pretreated with the voltage-sensitive K^+^ channel (K_V_) inhibitor 4-AP (1 × 10^−4^ mol/L), the inwardly rectifying K^+^ channel (K_IR_) inhibitor BaCl_2_ (1 × 10^−4^ mol/L), and the ATP-sensitive K^+^ channel (K_ATP_) inhibitor glibenclamide (1 × 10^−5^ mol/L), these three inhibitors failed to inhibit the vasodilatory effect induced by BSO, whereas the non-selective Ca^2+^-activated K^+^ channel (K_Ca_) inhibitor TEA (1 × 10^−3^ mol/L) significantly attenuated the diastolic effect of BSO (*p* < 0.001) (Figure 5B). This suggests that K_Ca_ activation may be involved in the BSO-induced vasodilatory response.

### 2.10. Effects of Cyclooxygenase Inhibitor on BSO-Induced Vasodilatory Effect

To determine whether PGI_2_ was involved in the vasodilatory effect induced by BSO, the pretreatment of endothelium-intact vessels with the cyclooxygenase inhibitor Indo (1 × 10^−5^ mol/L) failed to inhibit the vasodilatory effects of BSO (Figure 5C). This suggests that the PGI_2_-cAMP signaling pathway is not involved in the BSO-induced vasodilatory effect.

### 2.11. Effects of Muscarinic and Adrenergic Receptor Inhibitors on BSO-Induced Vasodilator Effect

To determine the influence of the autonomic nervous system on the vasodilatory effect of BSO, pretreatment of endothelium-intact vessels with the non-selective β-adrenergic receptor inhibitor propranolol (1 × 10^−6^ mol/L) and the muscarinic receptor inhibitor atropine (1 × 10^−6^ mol/L) had no effect on the vasodilatory effect of BSO (Figure 5D,E). This indicates that the vasodilatory effect induced by BSO is independent of the autonomic nervous system.

### 2.12. Effect of BSO on the Intensity of the Preconstructed Vasculature of PE in Calcium-Free Fluid

To further confirm the effect of BSO on endoplasmic reticulum calcium ions, the endothelium-denuded vascular rings in Ca^2+^-free Krebs solution with PE-induced vasoconstriction were tested, and the PE-induced constriction was significantly inhibited after incubation with BSO (100 µg/mL) (*p* < 0.01) (Figure 5F). This indicated that BSO could inhibit the release of intracytoplasmic Ca^2+^ mediated by the IP_3_ receptor system.

### 2.13. Effects of BSO on Systolic Blood Pressure and Heart Rate in SD Rats

The systolic blood pressure (SBP) (from 120.2 ± 3.5 to 104.8 ± 3.6 mmHg) and heart rate (from 435.0 ± 12.1 to 390.0 ± 14.2 beats/min) of rats were significantly decreased after 30s of femoral vein administration of a high dose of 5 mg/kg (Figure 6A,B), which was statistically significant (*p* < 0.05) compared to the values before the administration of BSO. The high, medium, and low dose groups of BSO also showed a certain degree of decrease in systolic blood pressure and heart rate, and the rate of change in systolic blood pressure and the heart rate of rats was statistically significant compared with that of the pre-administration group (Figure 6C,D) (*p* < 0.01).

## 3. Discussion

In the present study, the composition of BSO was determined using the HPLC-DAD and UPLC-MS methods; the results revealed that Diyu is rich in a variety of chemical components, the main components of which are amino acids, flavonoids, and alkaloids. These findings suggest that the n-butanol extract of Diyu may hold significant potential for pharmacological studies. Consequently, we next assessed its potential to induce vasodilatory effects.

The present study demonstrated that BSO exhibits a significant vasodilatory effect, with its mechanism of action primarily mediated through the endothelium-dependent eNOS-sGC-cGMP-K^+^ signaling pathway. Notably, it was further observed that BSO-induced vasodilation was associated with an increase in NO production, which occurred through the activation of eNOS phosphorylation via the PI3K/Akt and SOCE signaling pathways.

The PI3K/Akt signaling pathway has been reported to be located upstream of NO production [16]. eNOS is regulated by the endothelial PI3K/Akt signaling pathway [17,18]. Natural products have been shown to enhance eNOS expression [19,20]. In our study, we found that BSO increased Akt^ser473^, eNOS^ser1117^ phosphorylation, and NO production. These findings suggest that BSO activates the PI3K/Akt signaling pathway to upregulate eNOS expression for several reasons: BSO stimulates Akt phosphorylation, and the enhanced eNOS phosphorylation is almost completely blocked by PI3K/Akt inhibitors. Therefore, activation of the PI3K/Akt signaling pathway plays a crucial role in BSO-induced endothelial NO release. Additionally, the results indicated that the vasodilatory effect of BSO on PE-induced preconstriction was concentration-dependent, with the vasodilation being significantly inhibited upon the removal of the endothelium. This suggests an association with endothelial function. EDRFs, particularly NO, play a critical role in regulating endothelial and vascular smooth muscle tone [3]. Recent studies have demonstrated that NO enhances endothelial cell function and reduces inflammation in diabetic hypertensive rats [21]. To verify the effect of EDRFs, we used different inhibitors. The vasodilatory effect of BSO was inhibited by L-NAME but not by indomethacin, suggesting that BSO action is not related to the PGI_2_ pathway but rather to the activation of the NO pathway. Moreover, the results showed that ODQ significantly inhibited BSO-induced vasodilation. Furthermore, the Akt signaling inhibitor wortmannin significantly attenuated BSO-induced vasodilation, while BSO increased the phosphorylated Akt expression. These findings indicate that BSO-induced vasodilation is associated with the endothelium-dependent Akt-eNOS-NO-cGMP signaling pathway. Many vasodilators activate eNOS by increasing intracellular calcium ion concentration in the endothelial cells, thereby promoting vasodilation. This suggests that the role of Ca^2+^ in endothelial cells is critical for the process of NO synthesis and release [22].

In smooth muscle cells, two types of calcium channels are present: voltage-dependent calcium channels (VDCC) and receptor-operated calcium channels (ROCC). Vasoconstrictor phenylephrine (PE) and receptor binding induce the production of inositol trisphosphate (IP_3_) and the activation of protein kinase C (PKC) via diacylglycerol [23]. In this experiment, BSO-induced vasodilation was significantly inhibited in Ca^2+^-free Krebs solution. The results showed an association with Ca^2+^ in endothelial cells. Diltiazem pretreatment had no effect on BSO-induced vasodilation, indicating that the Ca^2^⁺ required for BSO-induced vasodilation does not influx through L-type channels but rather through other pathways. To further confirm the Ca^2^⁺-dependent vasodilatory effect of BSO, we examined the effects of the SOCE modulators TG, 2-APB, and Gd^3+^. The results showed that these SOCE modulators significantly inhibited the vasodilatory effect of BSO. This suggests that BSO exerts its physiological effects through the pathway SOCE-eNOS-NO. The addition of PE to a Ca^2^⁺-free Krebs solution caused a large release of Ca^2^⁺ from the endoplasmic reticulum via the IP3 pathway, leading to vascular ring contraction [24,25]. In vascular rings with the endothelium removed, BSO significantly inhibited PE-induced vasoconstriction in Ca^2^⁺-free Krebs, suggesting that BSO inhibits the release of intracellular Ca^2^⁺ mediated by the IP_3_ receptor system.

Potassium channels play a crucial role in regulating muscle contractile properties and vascular tone [26]. In smooth muscle cells, hyperpolarization of the membrane potential resulting from the opening of potassium ion channels is a key mechanism of arteriolar dilation [27]. Vasodilation induced by membrane hyperpolarization is attributed to enhanced potassium ion permeability [28]. Indeed, potassium ion channels in the vasculature indirectly influence vascular tone by altering the resting membrane potential. Through this pathway, these potassium channels not only help to maintain the resting membrane potential of vascular smooth muscle but also to modulate the vasculature’s diastolic and contractile responses. Many substances and drugs with vasoactive properties induce vasodilatory or contractile effects by opening or closing potassium ion channels [29]. In this experiment, pretreatment with the voltage-dependent K^+^ channel (K_V_) inhibitor 4-AP, the ATP-sensitive K^+^ channel (K_ATP_) inhibitor glibenclamide, and the inwardly rectifying K^+^ channel (K_IR_) inhibitor BaCl_2_ had no significant effect on BSO-induced vasodilation. However, pretreatment with the non-selective Ca^2^⁺-activated K⁺ channel (K_Ca_) inhibitor TEA significantly attenuated the vasodilatory effect of BSO on endothelium-intact vessels, suggesting that BSO-induced vasodilation is related to the activation of K_Ca_ channels. This mechanism may involve an increase in cGMP levels. It is currently believed that the downstream signaling pathway of cGMP involves the K⁺ channel. K_Ca_ channels can be activated by intracellular cGMP through the cGMP–protein kinase G signaling pathway, thereby inducing vasodilation [29]. In the present study, BSO induced significant hypotension and a decrease in heart rate in rats without causing a reflex increase in heart rate. These hypotensive effects may be related to the vasodilatory actions of BSO. A recent study reported that the ethanol extract of Diyu exhibited hypotensive and vasorelaxant effects [15]. Thus, although we demonstrated that BSO produced hypotension in normotensive rats, its beneficial effects on experimental hypertension and its clinical significance needs to be further investigated. Moreover, future studies should focus on the pharmacological mechanisms of individual compounds rather than the whole extract.

## 4. Materials and Methods

### 4.1. Chemicals and Reagents

Phenylephrine (PE), acetylcholine (Ach), levonitro-arginine methylester (N^G^-nitro-L-arginine-methylester, L-NAME), glibenclamide (Glib), ODQ (1H-[1,2,4]-oxadiazolo-[4,3-α]-quinoxalin-1-one), tetraethylammonium (TEA), indomethacin (Indo), diltiazem, atropine, thapsigargin (TG), propranolol, 2-aminoethyl diphenylborinate (2-APB), 4-aminopyridine (4-AP), barium chloride (BaCl_2_) and DMSO were purchased from Sigma Chemical Co., Ltd. (St. Louis, MO, USA). Gadolinium chloride (Gd^3+^), wortmannin, etc., were purchased from Biomol (Plymouth Meeting, PA, USA), The Krebs solution reagents, potassium chloride, sodium chloride, dextrose, magnesium chloride, calcium chloride, sodium bicarbonate, potassium phosphate, and magnesium sulfate, were purchased from Sigma Chemical Co., Ltd. The highest DMSO concentration (0.1%) was shown to have no effect on vascular tone in the control group.

### 4.2. Preparation of n-Butanol Extract of Diyu (BSO)

In this experiment, the dried roots of Diyu were extracted with 95% ethanol at room temperature three times, and the combined extracts were concentrated under a vacuum to obtain a crude extract. Petroleum ether, ethyl acetate, and *n*-butanol were used to extract the ethanol extract according to the order of polarity. Three different extraction layers were obtained. Finally, we selected the *n*-butanol extract for the vasodilation experiment. The experimental BSO was provided by Professor Gao Li from the College of Pharmacy, Yanbian University.

### 4.3. Experimental Animals

Healthy clean male Sprague Dawley (SD) rats weighing 250–300 g were provided by the Laboratory Animal Center of Yanbian University. The experiments were conducted in strict accordance with the regulations of the Animal Breeding Management and Use Committee. The subject animals were fed with standard block feed for one week and then experimented on. This research protocol has been approved by the Research Ethics Committee of the Animal Center of Yanbian University (Ethics number: YD20240311021).

### 4.4. HPLC-DAD and UPLC-MS Analysis

In our experiments, we used HPLC-DAD and UPLC-MS methods for the identification of BSO constituents. The BSO was weighed at 10 mg, dissolved in 1 mL of 30% methanol, vortexed for 2 min, ultrasonicated for 20 min, and filtered through a 0.22 µm membrane and placed in a vial for measurement. The sample handling methods for HPLC and LC-MS analysis included the Hitachi HPLC-DAD high-performance liquid chromatography (Japan) and the Thermo Fisher Scientific ultra-high-performance liquid chromatography system (Vanquish Flex) in tandem with a Q Exactive Orbitrap mass spectrometer (USA).

HPLC-DAD conditions: A Sepax Bio-C18 column (4.6 × 250 mm, 5 µm) was used for the separation of the BSO. The mobile phase was 0.1% formic acid in water (A) with gradient elution of acetonitrile (B). The gradient elution program was as follows: 0–15 min, 5% → 8% B; 15–50 min, 8%→16% B; 50–100 min, 16%→30% B; 100–105 min, 30%→95% B; 105–115 min, 95%→95% B; 115–120 min, 95%→5% B; and 120–130 min, 5%→5% B. The run time was 130 min, the flow rate was 1 mL/min, the injection volume was 10 μL, and the column temperature was 30 °C.

UPLC-MS conditions: An RP-C18 GP column (2.0 × 100 mm, 3 µm) was used for the separation of the BSO. The mobile phase was 0.1% formic acid in water (A) with a gradient elution of acetonitrile (B). The gradient elution program was as follows: 0–5 min, 2% B; 5–50 min, 2%→95% B; 50–55 min, 95%→95% B; 55–60 min, 95%→2% B; and 60–70 min, 2%→2% B. The flow rate was 0.3 mL/min, the injection volume was 2 μL, and the column temperature was 40 °C. The mass spectrometry conditions are shown in Table 2.

### 4.5. Determination of NO Production in Vascular Endothelial Cells

Primary cultured HUVECs were purchased from Gibco Cascade and contained 2.5% fetal bovine serum and growth supplements such as recombinant epidermal growth factor (rEGF), VEGF, human fibroblast growth factor-basic, ascorbic acid, human recombinant insulin-conjugated growth regulator, hydrocortisone, heparin, gentamicin, and amphotericin. To determine the production of NO in cells, HUVECs (2 × 10^5^ cells/well) in 6-well culture plates were pretreated with DAF-FM diacetate (ThermoFisher, Waltham, MA, USA) for 1 h. After removing the excess probe, they were treated with BSO for 30 min, and the fluorescence intensity was measured directly at Ex/Em 495/525 nm by a fluorescence spectrophotometer (Infinite F200 pro, TECAN, Männedorf, Switzerland) and examined under a fluorescence microscope (Eclipse Ti, Tokyo, Nikon).

### 4.6. Western Blot to Detect Changes in p-eNOS and p-Akt Protein Levels

The total proteins were extracted using a radioimmunoprecipitation assay buffer containing the Halt^TM^ protease inhibitor cocktail (ThermoFisher, OR, USA). Extracted proteins were quantified using the Bradford method and were electrophoretically transferred to nitrocellulose membranes (Bio-Rad Laboratories, Hercules, CA, USA) using 10% sodium dodecyl sulfate–polyacrylamide gel electrophoresis (SDS-PAGE). The membranes were blocked with 5% skim milk powder (Becton Dickinson, Le PontDe-Claix, France) containing 0.5% Tween 20–TBS for 1 h. The membranes were then incubated with antibodies against Akt and phosphorylated Akt, eNOS and phosphorylated eNOS, or β-actin (Biotechnology, CA, USA) at final dilutions of 1:1000 overnight at 4 °C. The secondary antibody was added for 1 h. The membrane was washed several times with 0.05% Tween 20. This was then detected by an enhanced chemiluminescence assay (Amersham Lab, Buckinghamshire, UK) program. This protein expression level was detected by analyzing the signals captured on nitrocellulose membranes (Millipore, MA, USA) using a ChemiDoc image analyzer (Bio-Rad Laboratories, Hertfordshire, UK).

### 4.7. Preparation of Isolated Thoracic Aortic Rings

After appropriate anesthesia and euthanasia, the removed thoracic aorta was transferred into Krebs–Henseleit (Krebs) solution containing a gas mixture pre-saturated with 95% O_2_ and 5% CO_2_ at 4 °C, which was composed of the following (mmol/L): NaCl 118.0, KCl 4.7, CaCl_2_ 1.5, KH_2_PO_4_ 1.2, NaHCO_3_ 25, MgSO_4_ 1.2, and Glucose 10, with the pH adjusted to 7.4 for HCl 1.2, NaHCO_3_ 25, MgSO_4_ 1.2, and Glucose 10, and the pH was adjusted to 7.4 for HCl. In Krebs solution, arterial connective and adipose tissues were removed, blood was removed, and vascular loops 3–4 mm in length were cut. In the endothelial injury model experiments, the inner wall of the vessel was scraped back and forth with a sharpened toothpick to remove the endothelial cells. The endothelial integrity was examined by first contracting the rings with PE at a concentration of 1 × 10^−6^ mol/L and then diastolizing the rings with Ach at a concentration of 1 × 10^−6^ mol/L. The endothelial integrity was also examined by using PE as a preconstrictor. The endothelium was considered to be removed if the diastolic rate of the vascular ring pre-contracted with PE and treated with Ach was less than 10%, and the endothelium was considered to be intact if it was more than 80%.

### 4.8. Determination of Vascular Ring Tone in Isolated Rat Thoracic Aorta

The bath contained Krebs solution passed through a mixture of 95% O_2_ and 5% CO_2_, with a pH of 7.4, at 37 °C. Two miniature hooks were placed symmetrically through the lumen of the vascular ring. The upper end of the vascular ring was attached to a tension transducer with a wire, and the lower end was fixed to the bottom of the bath so that the vascular ring was suspended horizontally in a 10 mL bath. The tension was increased by 0.25 g each time until the final tension was 1.0 g, then stabilized for 1 h, and the solution was changed every 10 min. The maximum contraction amplitude of the vascular ring induced by 1 × 10^−6^ mol/L of PE used was 100%, and this experiment responded to the change in vascular tone in terms of the diastolic rate, which is the percentage point of the ratio of the amplitude of vascular tone after the addition of the drug or the amplitude of the maximum contraction induced by PE.

### 4.9. Measurement of Rat Closing Blood Pressure and Heart Rate

SD rats (250–300 g) were anesthetized via an intraperitoneal injection of urethane at a dose of 1.2 g/kg. The right femoral artery and vein were exposed, and a catheter filled with heparin saline was inserted into the femoral artery and connected to a biological laboratory system (model BL-420S, Chengdu Taimeng, Chengdu, China). After blood pressure stabilization, BSO (1, 2.5, 5 mg/kg) was administered transvenously, and the blood pressure and heart rate before and after the administration were recorded. The vehicle indicates measurements obtained solely under urethane anesthesia.

### 4.10. Statistical Methods

The data of this experiment were analyzed and plotted with Graph Pad Prism TM 5.0 statistical software, and all of the data were expressed as the mean ± standard error (mean ± S.E.M). The comparison between the two groups of data was performed by a one-way ANOVA test. Additionally, the comparison between multiple groups of data was also performed by one-way ANOVA, and the statistical significance was defined as *p* < 0.05.

## 5. Conclusions

BSO-induced vasodilatory effects are mediated through an endothelium-dependent signaling pathway, which involves a two-step signaling pathway: first, the activation of the PI3K-Akt-eNOS-NO signaling pathway in vascular endothelial cells, followed by the activation of the NO-sGC-cGMP-K^+^ channel in vascular smooth muscle cells. Additionally, the present experimental data demonstrate that the vasodilatory effect of BSO is associated with the SOCE-eNOS pathway and the inhibition of Ca²⁺ mobilization from intracellular stores (Figure 7).

Natural drugs that can promote NO production have great potential for the treatment of cardiovascular diseases such as hypertension. In this study, we found that BSO exerts vasodilatory effects by promoting NO production, but that the effects of BSO on endothelial cells and smooth muscle cells need to be further investigated. The findings of this experiment provide a theoretical basis for the development of BSO as a new blood-pressure-lowering drug.

## Figures and Tables

**Figure 1 plants-14-01095-f001:**
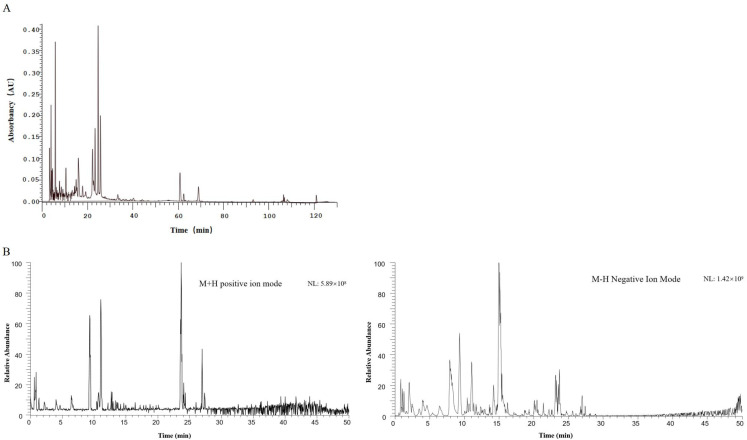
HPLC chromatogram (**A**) and total ion flow chromatogram (**B**) of n-butanol extract of Diyu.

**Figure 2 plants-14-01095-f002:**
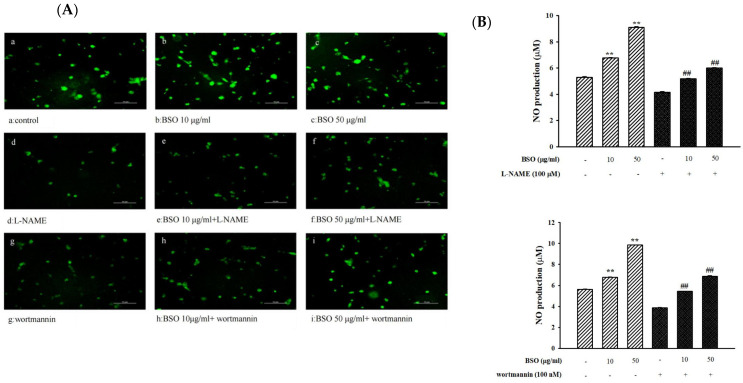
BSO stimulated the production of NO in vascular endothelial cells (**A**). HUVECs were pretreated with 100 μM L-NAME (**d**–**f**) or 0.1 μM wortmannin (**g**–**i**) and then stimulated with BSO. Control (**a**); 10 μg/mL BSO (**b**); 50 μg/mL BSO (**c**); L-NAME (**d**); L-NAME + 10 μg/mL BSO (**e**); L-NAME + 50 μg/mL BSO (**f**); wortmannin (**g**); wortmannin + 10 μg/mL BSO (**h**); and wortmannin + 50 μg/mL BSO (**i**). The fluorescence intensity of adherent HUVECs with L-NAME (**B**) and wortmannin (**c**) was measured by fluorescence microscopy. Values are the mean ± SE. ** *p* < 0.01 vs. control, ^##^ *p* < 0.01 vs. BSO.

**Figure 3 plants-14-01095-f003:**
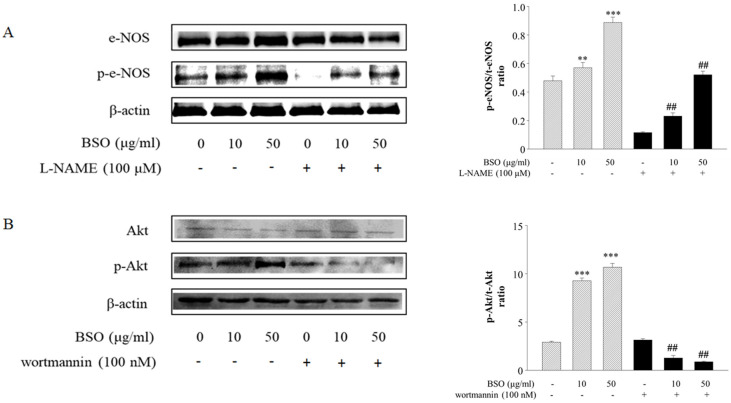
The effect of BSO on protein expression in vascular endothelial cells. BSO caused a concentration-dependent phosphorylation of eNOS (**A**) and Akt (**B**) in HUVECs. However, it was significantly inhibited by 100 μM L-NAME or 0.1 μM wortmannin, respectively. The data were expressed as means ± SD (n = 3). ** *p* < 0.01, *** *p* < 0.001 vs. control, ^##^ *p* < 0.01 vs. BSO.

**Figure 4 plants-14-01095-f004:**
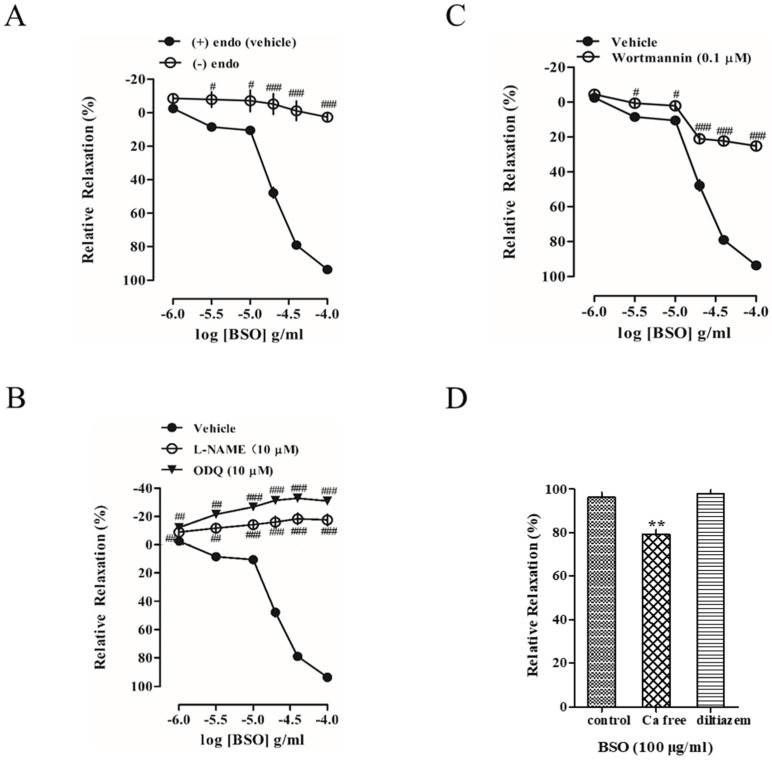
The effect of endothelial integrity (n = 5) and endothelial removal (n = 5) on the vasodilatory effect induced by BSO (**A**). Effects of L-NAME (10 μM, n = 4) and ODQ (10 μM, n = 4) on the BSO-induced vasorelaxation in the endothelium-intact aortic rings (**B**). Effect of wortmannin (0.1 μM, n = 5) on BSO-induced vasorelaxation in endothelium-intact aortic rings (**C**) (^#^ *p* < 0.05, ^##^ *p* < 0.01, ^###^ *p* < 0.001 vs. vehicle). Effects of Ca^2+^-free Krebs solution (n = 4) and diltiazem (n = 4) on vasodilation induced by BSO (**D**) (** *p* < 0.01 vs. control).

**Figure 5 plants-14-01095-f005:**
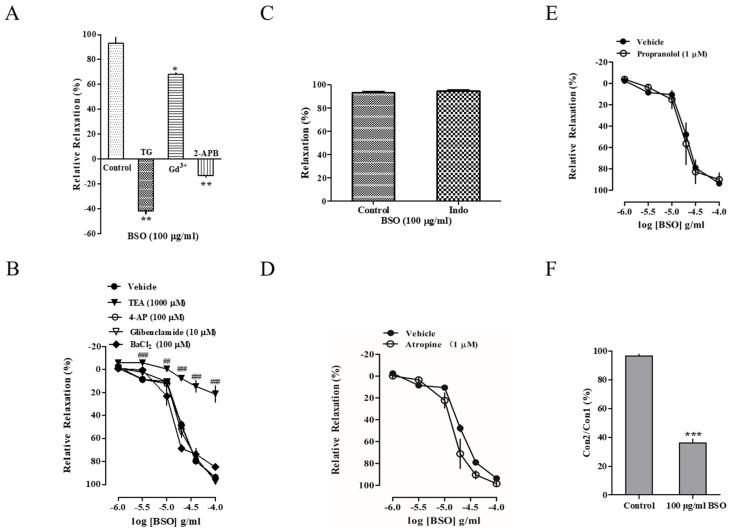
Effects of Ca^2+^ entry on BSO-induced vasorelaxation in endothelium-intact aortic rings. Effects of TG (n = 4), Gd^3+^ (n = 4), and 2-APB (n = 4) on vasodilation induced by BSO (**A**) (* *p* < 0.05, ** *p* < 0.01 vs. control). The effect of the ATP-sensitive K^+^ channel (K_ATP_) inhibitor glibenclamide (10 μM, n = 4), non-selective Ca^2+^-activated K^+^ channel (K_Ca_) inhibitor TEA (1000 μM, n = 4), voltage-dependent K^+^ channel (K_V_) inhibitor 4-AP (100 μM, n = 4), and inwardly rectifying K^+^ channel (K_IR_) inhibitor BaCl_2_ (100 µM, n = 4) on vasodilation induced by BSO in endothelium-intact aortic rings (**B**) (^##^ *p* < 0.01, ^###^ *p* < 0.001 vs. vehicle). The effect of cyclooxygenase inhibitor indomethacin (n = 4) on vasodilation induced by BSO in endothelium-intact aortic rings (**C**). Muscarinic and β-adrenergic receptor inhibitions of BSO-induced vasorelaxation in endothelium-intact aortic rings. Effect of atropine (**D**) (n = 4) and propranolol (**E**) (n = 4) on vasodilation induced by BSO. The inhibition of BSO on intracellular Ca^2+^ store release from SR-induced contraction by phenylephrine. The effect of BSO on the intensity of PE preconstriction in Ca^2+^-free solution (**F**) (*** *p* < 0.001 vs. vehicle).

**Figure 6 plants-14-01095-f006:**
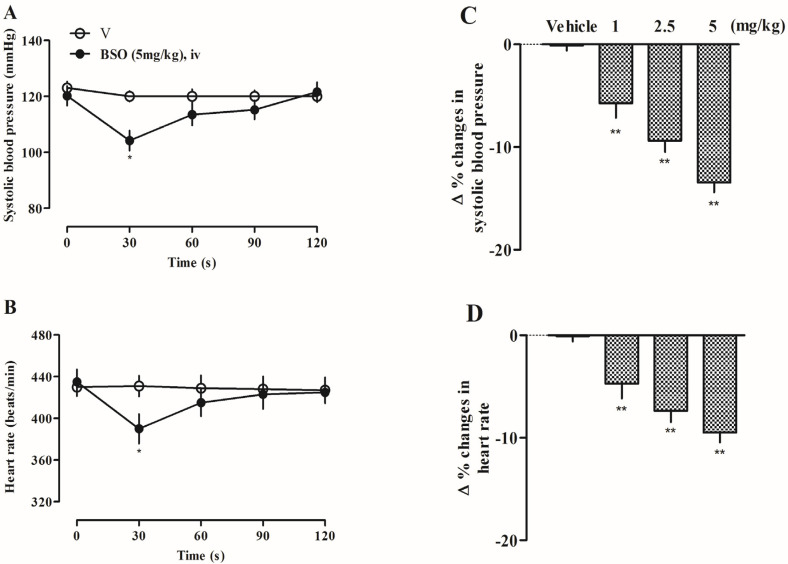
Effects of BSO on systolic blood pressure (**A**) (n = 4) and heart rate (**B**) (n = 4). (**C**) The rate of change in systolic blood pressure. (**D**) The rate of change in heart rate. (* *p* < 0.05, ** *p* < 0.01 vs. vehicle).

**Figure 7 plants-14-01095-f007:**
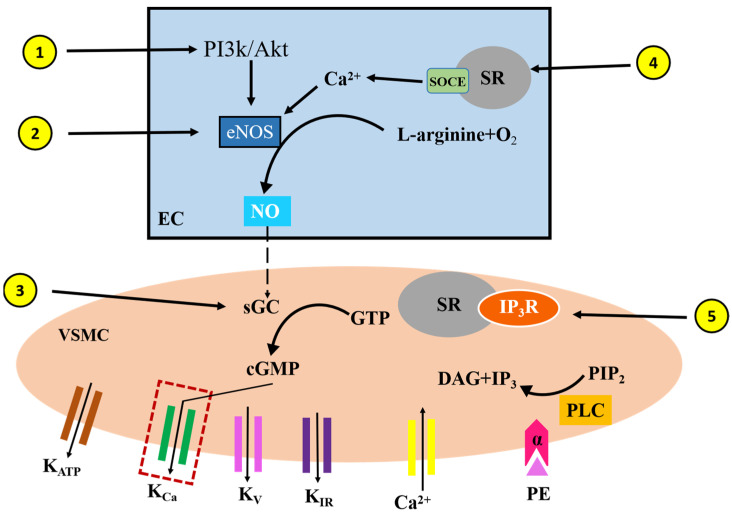
Schematic diagram of vasodilation induced by BSO. Steps 1 to 5 represent the signaling pathway through which BSO affects vascular relaxation. BSO induces vasorelaxation via endothelium-dependent signaling; its mechanisms are related to the activation of the PI3K-Akt-eNOS-NO signaling pathways in the endothelial cells and the activation of the NO-sGC-cGMP-K^+^ channel signaling pathways in the vascular smooth muscle cells. In addition, SOCE-eNOS pathways and the inhibition of Ca^2+^ mobilization from intracellular stores are associated with BSO-induced vasorelaxation.

**Table 1 plants-14-01095-t001:** Chemical constituents contained in *n*-butanol extract of Diyu as inferred by LC-MS.

Comp.	tR (min)	Calc. MW	Formula	ppm	Mode	MS^2^	Identification
**1**	0.67	174.11032	C_6_H_14_N_4_O_2_	−7.77	[M+H]+	116.07001, 72.93724, 70.06530, 55.93479	(2S)-2-amino-5-(diaminomethylideneamino)pentanoic acid
**2**	0.676	103.06291	C_4_H_9_NO_2_	−4.02	[M+H]+	87.04392, 69.03365, 60.08109, 58.06548	(2R)-2-aminobutanoic acid
**3**	0.814	180.06376	C_6_H_12_O_6_	2.09	[M−H]−	113.07639, 72.99217, 71.01303, 59.01295	(3R,4S,5R,6R)-6-(hydroxymethyl)oxane-2,3,4,5-tetrol
**4**	0.835	115.06268	C_5_H_9_NO_2_	−5.59	[M+H]+	71.06862, 70.06528, 68.04958, 58.06544	(2R)-pyrrolidine-2-carboxylic acid
**5**	0.861	342.11776	C_12_H_22_O_11_	4.53	[M−H]−	113.02408, 101.02386, 89.02372, 71.01302, 59.01295	α-D-Glucopyranosyl-α-D-glucopyranosid
**6**	0.903	290.12013	C_10_H_18_N_4_O_6_	−8.62	[M+H]+	175.23624, 130.08569, 112.08637, 70.06529	(2S)-2-[[N′-[(4S)-4-amino-4-carboxybutyl]carbamimidoyl]amino]butanedioic acid
**7**	1.94	267.09462	C_10_H_13_N_5_O_4_	−7.98	[M+H]+	153.01714, 136.06078, 119.03455, 92.02407, 79.01784	(2R,3R,4S,5R)-2-(6-aminopurin-9-yl)-5-(hydroxymethyl)oxolane-3,4-diol
**8**	2.479	170.02162	C_7_H_6_O_5_	0.57	[M−H]−	134.72574, 125.02406, 95.01319, 67.01817	3,4,5-trihydroxybenzoic acid
**9**	5.934	264.18163	C_15_H_24_N_2_O_2_	−8.14	[M+H]+	205.13214, 148.11089, 136.11107, 120.08011, 91.05392	(1R,2R,9S,17S)-13-oxido-7-aza-13-azoniatetracyclo [7.7.1.02,7.013,17]heptadecan-6-one
**10**	11.074	290.08044	C_15_H_14_O_6_	4.83	[M−H]−	203.07140, 159.04532, 123.04472, 109.02902, 83.01304	(2R,3S)-2-(3,4-dihydroxyphenyl)-3,4-dihydro-2H-chromene-3,5,7-triol
**11**	11.28	484.08754	C_20_H_20_O_14_	4.61	[M−H]−	271.04730, 211.02528, 169.01433, 125.02402, 107.01330	1,6-Bis-O-(3,4,5-trihydroxybenzoyl)hexopyranose
**12**	12.307	458.08107	C_22_H_18_O_11_	−8.38	[M+H]+	289.06839, 153.01712, 139.03793, 79.01785	[(2R,3R)-5,7-dihydroxy-2-(3,4,5-trihydroxyphenyl)-3,4-dihydro-2H-chromen-3-yl] 3,4,5-trihydroxybenzoate
**13**	12.908	450.11827	C_21_H_22_O_11_	4.57	[M−H]−	269.04678, 259.06226, 151.00346, 125.02404	(2R,3R)-7-[(2R,4R,5S,6R)-4,5-dihydroxy-6-(hydroxymethyl)oxan-2-yl]oxy-3,5-dihydroxy-2-(4-hydroxyphenyl)-2,3-dihydrochromen-4-one
**14**	13.401	197.11893	C_14_H_15_N	−7.71	[M+H]+	131.97343, 97.00727, 91.05400, 72.93723	N-benzyl-1-phenylmethanamine
**15**	13.725	304.05587	C_15_H_12_O_7_	−8	[M+H]+	231.06354, 153.01709, 123.04327, 65.03883	(2R,3R)-2-(3,4-dihydroxyphenyl)-3,5,7-trihydroxy-2,3-dihydrochromen-4-one
**16**	14.211	198.05324	C_9_H_10_O_5_	2.08	[M−H]−	166.99828, 123.00845, 95.01302, 67.01812	4-hydroxy-3,5-dimethoxybenzoic acid
**17**	14.566	788.11073	C_34_H_28_O_22_	4.45	[M−H]−	635.09155, 465.06940, 313.05804, 169.01428	1,2,3,6-Tetrakis-O-(3,4,5-trihydroxybenzoyl)-β-D-glucopyranose
**18**	14.965	302.00383	C_14_H_6_O_8_	−8.06	[M+H]+	257.00586, 201.01660, 145.01725, 89.03841	6,7,13,14-tetrahydroxy-2,9-dioxatetracyclo [6.6.2.04,16.011,15]hexadeca-1(15),4,6,8(16),11,13-hexaene-3,10-dione
**19**	16.239	428.17057	C_20_H_28_O_10_	5.43	[M+H]+	133.06563, 89.02371, 71.01303, 59.01295	(2E)-3-Phenyl-2-propen-1-yl 6-O-β-D-arabinofuranosyl-β-D-glucopyranoside
**20**	17.549	274.08183	C_15_H_14_O_5_	−8.35	[M+H]+	107.04868, 95.04886, 79.05424, 53.03899	3-(4-hydroxyphenyl)-1-(2,4,6-trihydroxyphenyl)propan-1-one
**21**	24.764	405.34206	C_22_H_47_NO_5_	−8.3	[M+H]+	300.28726, 256.26157, 135.00594, 70.06528	(2S,3S,5R,10R,12S,14S,15R,16R)-2-Amino-12,16-dimethyl-3,5,10,14,15-icosanepentol
**22**	24.955	255.25405	C_16_H_33_NO	−8.5	[M+H]+	212.23561, 93.03658, 88.07549, 53.00249	hexadecanamide
**23**	49	390.27373	C_24_H_38_O_4_	−8.41	[M+H]+	167.03261, 149.02219, 121.02760, 65.03879	dioctyl benzene-1,2-dicarboxylate

**Table 2 plants-14-01095-t002:** The mass spectrometry conditions.

Q-Orbitrap MS Conditions Polarity	Positive and Negative
AGC target	2 × 10^5^
Sheath gas flow rate	35 mL/min
Sweep gas flow rate	10 mL/min
Spray voltage	3.50 kV
Capillary temperature	320 °C
S-lens RF level	50.0

## Data Availability

The data used to support the findings of this study are included in the article.

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
