# Peer review of "Vasodilatory Effect of n-Butanol Extract from Sanguisorba officinalis L. and Its Mechanism"

_plants, 2025, doi:10.3390/plants14071095_

Round 1

Reviewer 1 Report

Comments and Suggestions for Authors

Vasodilatory effect and mechanism of n-butanol extract of Sanguisorba Officinalis L.

Although the effect of the butanol extract of S. officinalis apparently shows vasodilatory effect, the authors do not provide important details of the in vitro experiments. For example, the number of cells per well and whether the fluorescence readings were performed directly in the 6-well microplates, nor do they mention the wavelength at which these microplates were read. These details are important for the reproducibility of the experiments by other authors.

Regarding the animal experiments, they do not mention how the treated and control groups of animals were divided and how many animals were used per group.

I think there is an error, and if not, the fact that uracil was used to anesthetize the animals for the blood pressure and heartbeat experiments is wrong (p. 4 paragraph 183).

Cervical dislocation is not acceptable for rats over 200 grams of body weight due to their strong cervical musculature.

Author Response

I sincerely appreciate your valuable feedback on my paper. All the manuscripts have been carefully revised, and the modifications are highlighted in red in the main text.

Vasodilatory effect and mechanism of n-butanol extract of Sanguisorba Officinalis L.

Although the effect of the butanol extract of S. officinalis apparently shows vasodilatory effect, the authors do not provide important details of the in vitro experiments. For example, the number of cells per well and whether the fluorescence readings were performed directly in the 6-well microplates, nor do they mention the wavelength at which these microplates were read. These details are important for the reproducibility of the experiments by other authors.

  • We provided detail information.

Regarding the animal experiments, they do not mention how the treated and control groups of animals were divided and how many animals were used per group.

  • We corrected demonstration using HUVECs in Figure 3. In addition, it is indicated as n=3.

All animal experiments using the aortic ring as shown Figure 4, 5 were conducted in a tissue state rather than by administering the drug under cage housing conditions. In other words, this was an ex vivo experiment. Therefore, the control and treated groups are specified in the figure legends, with each group indicated as n=4.

The blood pressure and heart rate experiments presented in Figure 6 were conducted under controlled conditions, without drug administration during cage housing. Instead, the animals were briefly housed under identical conditions and subsequently anesthetized on the experimental table before BSO administration. Accordingly, the sample size for each group is indicated as n=4.

I think there is an error, and if not, the fact that uracil was used to anesthetize the animals for the blood pressure and heartbeat experiments is wrong (p. 4 paragraph 183).

  • We corrected it urethane at a dose of 1.2 g/kg.

Cervical dislocation is not acceptable for rats over 200 grams of body weight due to their strong cervical musculature.

  • In response to your comments, we have revised the statement to indicate that the experiment was conducted following appropriate anesthesia and euthanasia.

Reviewer 2 Report

Comments and Suggestions for Authors

In the present manuscript, the authors have studied the vasodilatory effects of n-butanol extract of Sanguisorba officinalis L. root (BSO) in Sprague-Dawley rats and in primary cultures of human umbilical vein endothelial cells (HUVEC) by analyzing nitric oxide (NO) production and the signaling pathways involved in the vasodilation mechanism.
The authors determined the composition of BSO by HPLC-DAD and UPLC-MS methods, showing that it is rich in amino acids, flavonoids and alkaloids and other chemical components. Using thoracic aorta sections and HUVEC as  experimental models, they demonstrated that BSO has a significant vasodilatory effect, through the endothelium-dependent eNOS-sGC-cGMP-K+ signaling pathway. In addition, they determined that BSO-induced vasodilation is associated with an increase in NO production through the activation of eNOS phosphorylation by the PI3K/Akt and SOCE signaling pathways. The work is interesting and the conclusions are relevant, however, some points need to be considered before accepting it for publication.

  • The wording of the abstract, introduction and results should be improved.
  • It would be convenient to incorporate and discuss the work of Jung J, Shin S, Park J, Lee K, Choi HY. Hypotensive and Vasorelaxant Effects of Sanguisorbae Radix Ethanol Extract in Spontaneously Hypertensive and Sprague Dawley Rats. Nutrients. 2023 Oct 24;15(21):4510. doi: 10.3390/nu15214510.

More detailed comments could be found in attached.

Comments on the Quality of English Language
  • The text has several syntax errors that make it difficult to read and understand, so it should be checked by a native English speaker.

Author Response

I sincerely appreciate your valuable feedback on my paper. All the manuscripts have been carefully revised, and the modifications are highlighted in red in the main text.

In the present manuscript, the authors have studied the vasodilatory effects of n-butanol extract of Sanguisorba officinalis L. root (BSO) in Sprague-Dawley rats and in primary cultures of human umbilical vein endothelial cells (HUVEC) by analyzing nitric oxide (NO) production and the signaling pathways involved in the vasodilation mechanism.
The authors determined the composition of BSO by HPLC-DAD and UPLC-MS methods, showing that it is rich in amino acids, flavonoids and alkaloids and other chemical components. Using thoracic aorta sections and HUVEC as experimental models, they demonstrated that BSO has a significant vasodilatory effect, through the endothelium-dependent eNOS-sGC-cGMP-K+ signaling pathway. In addition, they determined that BSO-induced vasodilation is associated with an increase in NO production through the activation of eNOS phosphorylation by the PI3K/Akt and SOCE signaling pathways. The work is interesting and the conclusions are relevant, however, some points need to be considered before accepting it for publication.

  • The wording of the abstract, introduction and results should be improved.
    • The manuscript was revised and errors were corrected with the assistance of a professional English editor.

  • It would be convenient to incorporate and discuss the work of Jung J, Shin S, Park J, Lee K, Choi HY. Hypotensive and Vasorelaxant Effects of Sanguisorbae Radix Ethanol Extract in Spontaneously Hypertensive and Sprague Dawley Rats. Nutrients. 2023 Oct 24;15(21):4510. doi: 10.3390/nu15214510.
    • We have cited the references you provided and incorporated them into the discussion. Thank you for your valuable advice

Reviewer 3 Report

Comments and Suggestions for Authors

In the abstract, all abbreviations (e.g., NOS, TEA) should be deciphered. Additionally, the conclusions are missing.

The introduction lacks sufficient information on the chemical composition of Sanguisorba officinalis L. Only groups of biologically active compounds (BACs) are briefly mentioned, but individual BACs exhibit pharmacological activity. Therefore, this section should be expanded and justified, explaining why the authors chose to study this specific activity based on literature data.

It is unclear why the butanol extract was selected. What happened to the petroleum ether and ethyl acetate fractions? Why were they not studied? The experimental design should be explained here, and the choice should be discussed.

What was the yield of the crude ethanol extract and the butanol fraction? It is necessary to describe how butanol was removed from the fraction. How were its residual amounts controlled? The butanol fraction should be characterized, including its appearance, certain quality parameters, quantitative characteristics, and the content of the main BAC groups.

Section 2.4. Information about the way of the substances identification should be added. Was their quantitative determination performed?

Lines 201–202. Adenosine is not an amino acid, and gallic acid is not a flavonoid. The description should be corrected in accordance with modern chemical classification.

Table 2. The chemical names of the compounds should be presented uniformly, either using trivial names or IUPAC names. Are there any data on the quantitative content of these substances? The discussion should highlight which of the identified compounds were dominant.

The discussion requires revision. Currently, the results of the chemical studies are not linked to the pharmacological findings. It is necessary to discuss the mechanisms of action based on individual compounds rather than the extract as a whole since BACs are responsible for the activity.

A conclusion section is missing and should be added.

Author Response

I sincerely appreciate your valuable feedback on my paper. All the manuscripts have been carefully revised, and the modifications are highlighted in red in the main text.

In the abstract, all abbreviations (e.g., NOS, TEA) should be deciphered. Additionally, the conclusions are missing.

  • We have revised the abbreviations in the abstract based on your advice. However, due to the presence of many lengthy terms, writing them in full would make the reading challenging. Therefore, we have modified them to be enclosed in parentheses for better readability.

The introduction lacks sufficient information on the chemical composition of Sanguisorba officinalis L. Only groups of biologically active compounds (BACs) are briefly mentioned, but individual BACs exhibit pharmacological activity. Therefore, this section should be expanded and justified, explaining why the authors chose to study this specific activity based on literature data.

  • We have cited the references you provided and incorporated them into the introduction and discussion. “Recently, it was reported that the ethanol extract of Diyu exhibits hypotensive and vaso-relaxant effects.” “Future studies should be conducted based on the pharmacological mechanisms of indi-vidual compounds rather than the whole extract.”

Jung, J.; Shin, S.; Park, J.; Lee, K.; Choi, H. Y. Hypotensive and Vasorelaxant Effects of Sanguisorbae Radix Ethanol Extract in Spontaneously Hypertensive and Sprague Dawley Rats. Nutrients 2023, 15, 4510.

It is unclear why the butanol extract was selected. What happened to the petroleum ether and ethyl acetate fractions? Why were they not studied? The experimental design should be explained here, and the choice should be discussed.

  • In the screening experiment to identify candidate compounds, the butanol fraction of Sanguisorba officinalis exhibited the highest vasodilation rate, and thus, the research was initiated focusing on this fraction.

What was the yield of the crude ethanol extract and the butanol fraction? It is necessary to describe how butanol was removed from the fraction. How were its residual amounts controlled? The butanol fraction should be characterized, including its appearance, certain quality parameters, quantitative characteristics, and the content of the main BAC groups.

  • A total of 31.85 g of extract was obtained from 200 g of dried medicinal herb using 1500 mL of butanol for fractionation. The yield was 93%.

Section 2.4. Information about the way of the substances identification should be added. Was their quantitative determination performed?

  • We added a summary of the two identification methods, and based on the limited data and conditions available at this time, we are not in a position to perform finer quantitative measurements.

Lines 201–202. Adenosine is not an amino acid, and gallic acid is not a flavonoid. The description should be corrected in accordance with modern chemical classification.

  • We are very sorry for our careless mistake, thank you for the reminder and it has now been corrected in accordance with the modernised scientific taxonomy.

Table 2. The chemical names of the compounds should be presented uniformly, either using trivial names or IUPAC names. Are there any data on the quantitative content of these substances? The discussion should highlight which of the identified compounds were dominant.

  • Thank you for your careful examination, in our revised manuscript, the compound names in Table 2 have been uniformly used as IUPAC names, a portion of previously obtained data has been added to Table 2, and summaries have been added to the Discussion and Conclusion sections to make the article more complete. We think it is a very good suggestion to discuss the mechanism of action of individual compounds, but at present, due to the limitation of experimental conditions, we are more interested in doing pharmacological research, so we only made the whole extract of Diyu without finer extraction and identification, we sincerely thank you for your valuable comments, and if the experimental conditions are ripe at a later stage, we will surely carry out the relevant research according to your suggestions!

The discussion requires revision. Currently, the results of the chemical studies are not linked to the pharmacological findings. It is necessary to discuss the mechanisms of action based on individual compounds rather than the extract as a whole since BACs are responsible for the activity.

  • We discussed it. “In addition, future studies should be conducted based on the pharmacological mechanisms of individual compounds rather than the whole extract.”

A conclusion section is missing and should be added.

  • We added it.

Round 2

Reviewer 2 Report

Comments and Suggestions for Authors

In the new version of the manuscript the authors made all the suggestions of thi reviewer. The manuscript has been significantly improved, so I recommend its publication in Plants.

Author Response

We sincerely appreciate your valuable feedback on our paper. All the manuscripts have been carefully revised, and the modifications are highlighted in red in the main text.

Reviewer 3 Report

Comments and Suggestions for Authors

Most of recommendations were ignored.

Author Response

(The authors gave the same response as above.)

Round 3

Reviewer 3 Report

Comments and Suggestions for Authors

No comments